# Learning Context with Priors for 3D Interacting Hand-Object Pose Estimation

## ABSTRACT

Achieving 3D hand-object pose estimation in interaction scenarios is challenging due to the severe occlusion generated during the interaction. Existing methods address this issue by utilizing the correlation between the hand and object poses as additional cues. They usually first extract the hand and object features from their respective regions and then refine them with each other. However, this paradigm disregards the role of a broad range of image context. To address this problem, we propose a novel and robust approach that learns a broad range of context by imposing priors. First, we build this approach using stacked transformer decoder layers. These layers are required for extracting image-wide context and regional hand or object features by constraining cross-attention operations. We share the context decoder layer parameters between the hand and object pose estimations to avoid interference in the context-learning process. This imposes a prior, indicating that the hand and object are mutually the most important context for each other, significantly enhancing the robustness of obtained context features. Second, since they play different roles, we provide customized feature maps for the context, hand, and object decoder layers. This strategy facilitates the disentanglement of these layers, reducing the feature learning complexity. Finally, we conduct extensive experiments on the popular HO3D and Dex-YCB databases. The experimental results indicate that our method significantly outperforms state-of-the-art approaches and can be applied to other hand pose estimation tasks. The code will be released.

## CCS CONCEPTS

• **Computing methodologies** → **Computer vision tasks**.

## KEYWORDS

Hand-Object Pose Estimation, Transformer

## 1 INTRODUCTION

The 3D hand-object pose estimation task simultaneously estimates the hand and object's poses in interaction scenarios. It has been widely applied in augmented and virtual reality [21, 59, 68, 79], human-computer interaction [48, 57, 73, 86], robotic manipulation [22], robot-assisted surgeries [69], and embodied artificial intelligence [37, 72]. While significant progress has been made in both 3D hand and object pose estimation [4, 5, 14–16, 18, 30, 55,

Permission to make digital or hard copies of all or part of this work for personal or classroom use is granted without fee provided that copies are not made or distributed for profit or commercial advantage and that copies bear this notice and the full citation on the first page. Copyrights for components of this work owned by others than the author(s) must be honored. Abstracting with credit is permitted. To copy otherwise, or republish, to post on servers or to redistribute to lists, requires prior specific permission and/or a fee. Request permissions from permissions@acm.org.

*ACM MM, 2024, Melbourne, Australia*

© 2024 Copyright held by the owner/author(s). Publication rights licensed to ACM.
ACM ISBN 978-x-xxxx-xxxx-x/YY/MM
https://doi.org/10.1145/nnnnnnn.nnnnnnn

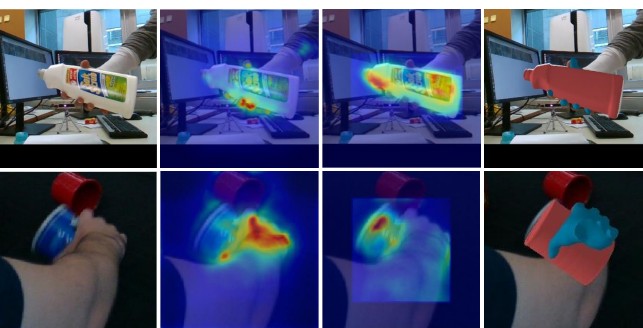

Figure 1: Illustration of attention maps produced by the context decoder layer in our Learning Context with Priors (LCP) framework. The four columns show the original images, attention maps for hand and object queries, and final pose estimation results. The hand attention maps cover the hand, object, and forearm. Meanwhile, the object attention maps highlight the object and the hand touching the object.

61, 66, 70, 71, 74, 85], jointly estimating 3D hand-object poses remains challenging. This is primarily due to the mutual occlusion between the interacting hand and object, as illustrated in Fig.1. Occlusion leads to information loss and interference, significantly affecting the pose estimation accuracy. It is important to overcome this challenge for applications like robotic manipulation [22] and robot-assisted surgeries [69], as inaccurate pose estimation may lead to safety issues.

Since the hand and object poses are coupled when grasping occurs, their pose correlation is a valuable cue that relieves occlusion. Generally, existing approaches [45, 47] extract the hand and object features from their respective regions and then enhance each other using transformer-like modules [65]. These enhanced features are used to predict 3D hand and object poses independently. Although this paradigm promotes 3D hand-object pose estimation, it overlooks leveraging a broad range of image context, such as the pose correlation between the hand and arm (see Fig. 1).

However, exploring a broad range of context for this task is challenging, as the model may struggle to identify useful cues in the entire image. Herein, we solve this problem by imposing priors in the context-learning process. First, our proposed approach utilizes the transformer decoder layers [2]. These layers are required for extracting the image-wide context, the fine-grained hand, and the object features by restricting the cross-attention operation scope. The disentangling operation relieves the latter two features from interference outside the hand or object regions. More importantly, we propose sharing the context decoder layer parameters between the 3D hand and object pose estimation tasks. This imposes a strong prior, emphasizing that the interacting hand and object are mutually the most important context for each other. This strategy enables extracting context features more robustly. Moreover, the obtained

context features are transferred to the hand and object decoder layers as decoder embeddings. In this way, we fuse robust image-wide context and fine-grained hand or object features for 3D hand-object pose estimation purposes.

Second, since they play different roles, we provide customized feature maps for the context, hand, and object decoder layers, facilitating disentanglement between them. However, since the context and hand (or object) layers are in series as stacked decoder layers, the key to this approach is to ensure that the feature maps utilized are in similar feature spaces. Accordingly, we adopt the backbone proposed in [45] that not only disentangles the hand and object feature maps but also ensures them are in the same feature space. Then, the two types of feature maps are fed into the hand and object decoder layers, respectively. Furthermore, we propose concatenating the above hand and object feature maps along the channel dimension and then halving the channel number using an efficient $1 \times 1$ convolution layer. Then, the obtained feature maps are utilized as the value and key for the context decoder layer. The experimentation section shows that this operation significantly enhances the 3D hand-object pose estimation performance.

To demonstrate the effectiveness of our approach, we conducted extensive experiments on two widely used databases: HO3D [19] and Dex-YCB [3]. The experimental results validate the effectiveness of each key design and show that our method consistently outperforms state-of-the-art approaches. Moreover, our method can also be applied to the 3D interacting-hands pose estimation, showcasing exceptional performance.

## 2 RELATED WORK

**3D Hand-Object Pose Estimation.** Existing methods for this task typically employ a parametric hand model (e.g., MANO [32]) and assume that the 3D object models are available. This enables them to focus on the 3D hand and object pose prediction. Furthermore, another line of closely related approaches [6–8, 24, 29, 33–35, 41, 63, 77, 78, 83] called 3D hand-object reconstruction exists. This approach does not assume the availability of the 3D object models; instead, it focuses on reconstructing the hand and object meshes. This paper targets the 3D hand-object pose estimation task with a single RGB image adopted as the input data. This section reviews existing research from model architecture, optimization strategy, and training data perspectives.

For model architecture, previous approaches [11, 62] adopt a shared encoder and decoder for the hand and object pose estimation. However, they struggle to thoroughly explore the hand and object's unique characteristics. To address this problem, the following works [20, 23, 45, 47] adopt a shared encoder and separate decoders for the two sub-tasks. Given the strong correlation between the poses of an interacting hand and object, recent methods [20, 45, 47] usually utilize the hand and object features to enhance each other. First, they extract the hand and object features from their respective regions. Then, they use them to enhance each other using transformer-like modules [65]. For example, Hampali et al. [20] first detect 2D hand and object keypoints. Then, they extract each keypoint's features. In addition, Liu [47] and Lin et al. [45] extract the hand and object features from their respective bounding boxes. Lin et al. [45] devise a novel backbone model that disentangles the

hand and object feature maps and ensures they are in a similar feature space, which facilitates the mutual enhancement between the hand and object features.

Model optimization strategies typically utilize physically plausible constraints to refine the hand-object poses, particularly focusing on stable contact between the interacting hand and object. For example, some studies [1, 23] introduce the contact and repulsion loss functions that encourage stable contact and discourage repulsion between the hand-object surfaces. Also, some methods [17, 38, 64, 76] explicitly model the hand-object contact. For instance, Grady et al. [17] and Tse et al. [64] first estimate contact maps between the hand and object; then, the maps are used to optimize the hand poses accordingly. Another noteworthy approach [76] utilizes Contact Potential Fields (CPF) to refine the predicted hand mesh with attractive and repulsive energy terms between the hand-object meshes.

The third category of methods addresses the difficulty on data annotation. For example, some studies [1] utilize multiple visual cues, such as the object detection results, 2D hand pose estimation, object instance segmentation, and hand-object contact area, to estimate 3D hand-object poses without supervision. This strategy does not require ground truth annotations. Some studies employ constraints based on temporal consistency to propagate the labels from sparsely annotated frames to the unannotated ones [23] or to enhance the quality of pseudo-labels [47] estimated by one off-the-shelf algorithm [47]. Finally, Yang et al. [40] introduce an online data augmentation method that synthesizes training data with free labels to promote data diversity.

In this paper, we explore model architecture. Compared to existing research on 3D hand-object pose estimation, we propose a novel approach that robustly leverages both image-wide context and fine-grained regional features.

**Transformer-based Pose Estimation.** Detection Transformer (DETR) [2] and its variants [52, 87] utilize decoder layers and object queries to aggregate image-wide context efficiently for the object detection task. DETR has been applied to various pose estimation tasks, e.g., the 2D and 3D human pose estimation task [9, 12, 31, 39, 46, 51, 75, 80]. For instance, some approaches [39, 46, 51, 75] redefine the 2D human pose estimation task as a regression problem. Unlike traditional heatmap-based methods [60, 67, 84], they utilize a query to predict one specific human keypoint. Similarly, other approaches [9, 12, 80] employ a query to predict the 3D coordinates of one specific human-body joint or vertex. Considering the increased number of vertices and joints, they design a series of methods to reduce model size and computational cost. For example, Huang et al. [31] use a limited number of queries to predict the pose and shape parameters of the human SMPL [49] model.

In contrast to the above human pose estimation tasks, we discovered that aggregating image-wide context for hand-object pose estimation, which is more fine-grained and vulnerable to context inferences, is more challenging. This paper addresses this problem by imposing strong priors in the context-learning process.

## 3 METHODS

In this section, we first provide an overview to the 3D hand-object pose estimation problem and our Learning Context with Priors (LCP) framework in Section 3.1. Then, we introduce LCP's main

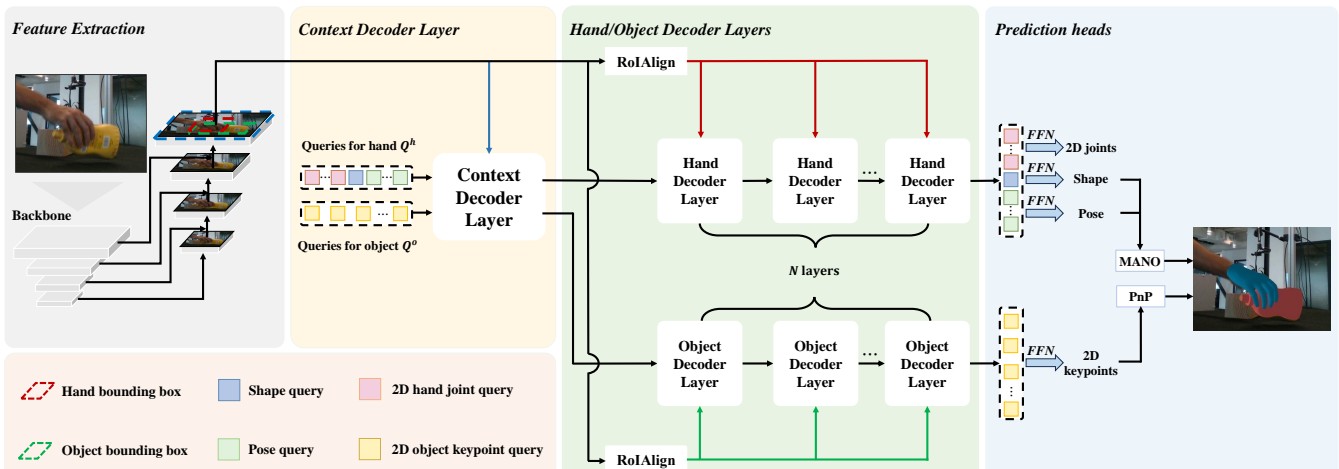

**Figure 2: An overview of our LCP framework. It stacks the context, hand, and object transformer decoder layers on the feature maps produced by the backbone. The same context decoder layer is adopted for the hand and object pose estimation tasks. This implicitly imposes a prior that the interacting hand and object are mutually the most important context for each other. The hand and object decoder layers extract fine-grained features from their ROI regions, respectively. The context and hand decoder layers share queries for hand pose estimation. Meanwhile, the context and object decoder layers share queries for object pose estimation.**

components in Section 3.2 and Section 3.3 using the ResNet-50-FPN model as backbone. Finally, we enhance the LCP's capacity using a stronger backbone in Section 3.4.

## 3.1 Overview

We adopt the popular MANO model [32] $\mathcal{M}(\boldsymbol{\theta}, \boldsymbol{\beta})$ to represent the hand, where $\boldsymbol{\theta} \in \mathbb{R}^{16 \times 3}$ and $\boldsymbol{\beta} \in \mathbb{R}^{10}$ represent the hand pose and shape coefficients, respectively. Meanwhile, the object mesh is assumed available, so we focus on object pose estimation, including the rotation $\mathbf{R} \in SO(3)$ and translation $\mathbf{T} \in \mathbb{R}^3$ parameters [71]. Similar to existing studies [45, 47], instead of directly regressing $\mathbf{R}$ and $\mathbf{T}$, we first estimate the 2D coordinates of object keypoints. Then, we employ the Perspective-n-Point (PnP) algorithm [36] to calculate the object pose. The goal of 3D hand-object pose estimation is to estimate $\boldsymbol{\theta}, \boldsymbol{\beta}, \mathbf{R}$, and $\mathbf{T}$, using a single unified framework with an RGB image as input.

The overview of LCP is provided in Fig. 2. We feed a single RGB image $\mathbf{I} \in \mathbb{R}^{H \times W \times 3}$ into the backbone, which consists of a ResNet-50 model [26] and a Feature Pyramid Network (FPN) [44]. The FPN fuses four feature map scales from the ResNet-50 model and outputs $\mathbf{F} \in \mathbb{R}^{H/4 \times W/4 \times C}$, where $C$ denotes the channel number. Beside the backbone, LCP mainly consists of three components: the context, hand, and object decoder layers.

The context, hand, and object decoder layers play different roles. They are required to extract the image-wide context, fine-grained hand, and object features, by constraining the scope of cross-attention operations. This disentanglement operation can free the latter two features from interference contained in the context. As illustrated in Fig. 2, the context and hand (or object) decoder layers are in series as stacked layers. The obtained context features are transferred to the hand and object decoder layers as decoder embeddings. The

key is how to obtain the context features in a robust manner for the hand and object, respectively.

## 3.2 Learning Context with Priors

Naively employing the cross-attention operation to search for image-wide context is suboptimal, as the model may struggle to identify useful cues and be vulnerable to inferences. We address this problem by imposing priors for the hand and object, indicating that the interacting hand and object are mutually the most important context for each other. Moreover, the hand is flexible, and its pose is not only related to the interacting object but also to the wrist and forearm poses. In contrast, the object is rigid; therefore, its pose correlates with that of the hand only when the hand touches the object [45, 47], as shown in Fig. 1.

Based on the above observation, we propose sharing the context decoder layers for the hand and object pose estimation tasks. This imposes a strong prior that the hand and object are mutually the most important context for each other. Specifically, we provide the context decoder layers with one group of learnable queries $\mathbf{Q}^h \in \mathbb{R}^{N_h \times C}$ for the hand and another for the object $\mathbf{Q}^o \in \mathbb{R}^{N_o \times C}$, where $N_h$ and $N_o$ denote the number of queries for the hand and object, respectively. Then, the context decoder layers transform $\mathbf{Q}^h$ and $\mathbf{Q}^o$ into a set of decoder embeddings $\mathbf{D}_c^h \in \mathbb{R}^{N_h \times C}$ and $\mathbf{D}_c^o \in \mathbb{R}^{N_o \times C}$, respectively. The meanings of the hand and object queries are described in Section 3.3. The above process is formulated as follows:

$$\mathbf{D}_c^h = f_c(\mathbf{Q}^h, \mathbf{D}_0^h, \mathbf{F}, \mathbf{E}_{pos}), \quad (1)$$

$$\mathbf{D}_c^o = f_c(\mathbf{Q}^o, \mathbf{D}_0^o, \mathbf{F} \odot \mathbf{M}^o, \mathbf{E}_{pos}), \quad (2)$$

where $f_c(.,.,.,.)$ denotes the stacked context decoder layers. $\mathbf{D}_0^h$ and $\mathbf{D}_0^o$ denote the initial decoder embeddings (i.e., zero vectors). $\mathbf{D}_c^h$ and $\mathbf{D}_c^o$ represent the output decoder embeddings by the context decoder layers. $\mathbf{E_{pos}} \in \mathbb{R}^{(H/4 \times W/4) \times C}$ stands for the position encoding. The matrix $\mathbf{M}^o$ is a mask and $\odot$ executes the Hadamard Product between $\mathbf{M}^o$ and each channel of $\mathbf{F}$. Its purpose is to constrain the cross-attention operation for $\mathbf{Q}^o$ within the object bounding box.

### 3.3 Hand and Object Decoder Layers

The hand and object decoder layers use $\mathbf{D}_c^h$ and $\mathbf{D}_c^o$ as initial decoder embeddings, respectively. Furthermore, these layers continue to employ $\mathbf{Q}^h$ and $\mathbf{Q}^o$ as queries. Compared to the context decoder layers, they focus on extracting fine-grained features from the hand and object regions, respectively. This is described as follows:

$$\mathbf{D}_{cl}^h = f_h(\mathbf{Q}^h, \mathbf{D}_c^h, \mathbf{F}^h, \mathbf{E}_{pos}^h), \tag{3}$$

$$\mathbf{D}_{cl}^o = f_o(\mathbf{Q}^o, \mathbf{D}_c^o, \mathbf{F}^o, \mathbf{E}_{pos}^o), \tag{4}$$

where $f_h(.,.,.,.)$ and $f_o(.,.,.,.)$ denote the stacked hand and object decoder layers, respectively. $\mathbf{D}_{cl}^h$ and $\mathbf{D}_{cl}^o$ stand for the final decoder embeddings for the hand and object pose estimation tasks. They contain both image-wide context features and fine-grained hand or object features. Similar to existing studies [45, 47], $\mathbf{F}^h \in \mathbb{R}^{32 \times 32 \times C}$ and $\mathbf{F}^o \in \mathbb{R}^{32 \times 32 \times C}$ are obtained from $\mathbf{F}$ using the ROIAlign [25] operation, according to the hand and object bounding boxes, respectively. Finally, $\mathbf{E}_{pos}^h$ and $\mathbf{E}_{pos}^o$ represent the positional embeddings for $\mathbf{F}^h$ and $\mathbf{F}^o$, respectively.

Notably, $\mathbf{Q}^h$ includes three types of highly correlated hand queries. The queries are utilized to predict the hand pose $\boldsymbol{\theta} \in \mathbb{R}^{16 \times 3}$ and shape $\boldsymbol{\beta} \in \mathbb{R}^{10}$ of the MANO model [32] and 2D hand joint coordinates $\mathbf{J}^h \in \mathbb{R}^{21 \times 2}$. Accordingly, we adopt 16 pose queries to predict each of the 3D joint angles in the hand kinematic structure, where each 3D joint angle is represented by three parameters. Furthermore, we utilize an extra query for estimating $\boldsymbol{\beta}$. Additionally, we add 21 more queries, each used to predict the 2D coordinates of one hand joint. These three types of hand queries are adopted in each context and hand decoder layer. The decoder embeddings obtained from these queries exchange information through the self-attention blocks in the decoder layers.

In addition, $\mathbf{Q}^o$ incorporates a single type of queries for objects. It includes 21 unique queries, each used to predict the 2D coordinates of a specific object keypoint. The same as [45, 47], the 21 keypoints include the projection of 8 corner points, 12 midpoints along the edges, and the central point of the 3D object bounding box onto the 2D image plane. Like the hand decoder layers, the decoder embeddings extracted by the 21 object keypoint queries conduct self-attention operations to capture the relationships in object structures.

Finally, the decoder embeddings obtained by $\mathbf{Q}^h$ and $\mathbf{Q}^o$ make predictions via respective feed-forward networks (FFNs). In the training stage, we supervise the outputs of each context, hand, and object decoder layer. In the testing stage, we use the $\boldsymbol{\theta}$ and $\boldsymbol{\beta}$ coefficients predicted by the final hand decoder layer to represent the hand pose. Regarding the object, we follow existing works [45, 47] and adopt the PnP [36] algorithm to calculate the final object

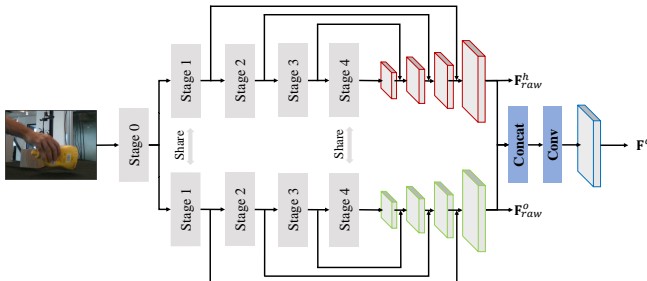

**Figure 3: Illustration of the CFM backbone.**

pose according to the object keypoints predicted by the final object decoder layer.

### 3.4 LCP with Customized Feature Maps

The LCP model in the above subsections is based on the ResNet-50-FPN backbone. However, our context, hand, and object decoder layers play different roles; therefore, it is more reasonable to provide them with customized feature maps (CFM), facilitating the disentanglement between these layers. Moreover, since the context layer and the hand (or object) layers are stacked in series, it is essential to ensure that these customized feature maps are still in similar feature spaces.

We achieve this goal with the help of the backbone proposed in [45], which disentangles the hand and object feature maps and ensures that they share the same feature space. However, this backbone's output comprises only the hand and object feature maps. The two feature maps are denoted as $\mathbf{F}_{raw}^h \in \mathbb{R}^{H/4 \times W/4 \times C}$ and $\mathbf{F}_{raw}^o \in \mathbb{R}^{H/4 \times W/4 \times C}$, respectively. To integrate this backbone with LCP, we concatenate $\mathbf{F}_{raw}^h$ and $\mathbf{F}_{raw}^o$ along the channel dimension and then halve the channel number by an efficient $1 \times 1$ convolution layer. The obtained feature maps $\mathbf{F}^c \in \mathbb{R}^{H/4 \times W/4 \times C}$ are utilized as the value and key for the context decoder layer, while $\mathbf{F}^h$ and $\mathbf{F}^o$ are obtained using the ROIAlign [25] operation from $\mathbf{F}_{raw}^h$ and $\mathbf{F}_{raw}^o$, respectively. Finally, we employ the customized feature maps $\mathbf{F}^c$, $\mathbf{F}^h$, and $\mathbf{F}^o$ to our LCP model's context, hand, and object layers, respectively. The above process to obtain the customized feature maps is illustrated in Fig. 3.

### 3.5 Loss Functions

The LCP's total loss comprises two parts: one for the hand and another for the object. It can be formulated as follows:

$$\mathcal{L}_{total} = \mathcal{L}_{hand} + \mathcal{L}_{object}. \tag{5}$$

The details of the two parts are as follows:

$$\mathcal{L}_{hand} = \lambda_{\boldsymbol{\theta}} \left\| \boldsymbol{\theta} - \hat{\boldsymbol{\theta}} \right\|_2 + \lambda_{\boldsymbol{\beta}} \left\| \boldsymbol{\beta} - \hat{\boldsymbol{\beta}} \right\|_2 + \lambda_{\mathbf{V}} \left\| \mathbf{V} - \hat{\mathbf{V}} \right\|_2 + \\ \lambda_{\mathbf{J}} \left\| \mathbf{J} - \hat{\mathbf{J}} \right\|_2 + \lambda_{\mathbf{J}^h} \left\| \mathbf{J}^h - \hat{\mathbf{J}}^h \right\|_1, \tag{6}$$

$$\mathcal{L}_{object} = \lambda_{\mathbf{J}^o} \left\| \mathbf{J}^o - \hat{\mathbf{J}}^o \right\|_1. \tag{7}$$

where $\lambda_{\boldsymbol{\theta}}$, $\lambda_{\boldsymbol{\beta}}$, $\lambda_{\mathbf{V}}$, $\lambda_{\mathbf{J}}$, $\lambda_{\mathbf{J}^h}$, and $\lambda_{\mathbf{J}^o}$ are set as 10, 0.1, 10000, 10000, 250, and 300, respectively. $\boldsymbol{\theta}$ and $\boldsymbol{\beta}$ represent the ground truth MANO [32] pose and shape coefficients, respectively. In addition, $\mathbf{V}$

and $\mathbf{J}$ denote the ground truth 3D coordinates of the hand vertices and joints. They are obtained according to the MANO [32] model with ground-truth $\boldsymbol{\theta}$ and $\boldsymbol{\beta}$ coefficients. Furthermore, $\mathbf{J}^h$ and $\mathbf{J}^o$ represent the 2D coordinates of hand joints and object keypoints, respectively. Moreover, $\hat{}$ denotes the predicted values. We impose the L2 loss function on $\boldsymbol{\theta}$, $\boldsymbol{\beta}$, $\mathbf{V}$, and $\mathbf{J}$, and adopt the L1 loss for $\mathbf{J}^h$ and $\mathbf{J}^o$.

## 4 EXPERIMENT

### 4.1 Datasets and Metrics

**HO3D**. The HO3D database [19] comprises data from 10 subjects executing various interactive actions with 10 objects. According to the official protocol, HO3D includes a total of 66,034 training and 11,524 testing images.

**Dex-YCB**. Dex-YCB [3] is a recently released large-scale hand-object manipulation dataset. Its images were captured under more challenging circumstances. It contains 582,000 frames sampled from over 1,000 video sequences. These sequences record 10 subjects interacting with 20 objects. We use the official s0 splitting protocol to divide the dataset into training and testing sets.

**InterHand2.6M**. InterHand2.6M [54] is a popular 3D interacting-hands pose estimation database. Its images include various interaction scenarios between two hands. It contains 1.36 million training and 849,000 testing images.

We adopt two popular metrics to evaluate the hand pose estimation performance for HO3D and Dex-YCB. Specifically, these metrics include the mean per joint position error with procrustes alignment (PA-MPJPE) and the mean per joint position error without procrustes alignment (MPJPE). MPJPE measures the average Euclidean distance between the predicted coordinate of each hand joint and the ground truth in millimeters (mm), while PA-MPJPE corrects the MPJPE score using procrustes analysis.

Similar to existing studies [45, 47], we evaluate the object pose estimation performance only for the objects viewed during training, using the average distance of model points (ADD) as the metric. The ADD metric [27] assesses whether the average deviation of the predicted model points is within 10% of the object's diameter. Consistent with the approach in [45], we adopt a symmetric version of the ADD metric, i.e., ADD(-S) [27, 28], on the Dex-YCB dataset [3], as some objects in this database are symmetric.

### 4.2 Implementation Details

All the experiments are conducted using PyTorch [56] and NVIDIA GeForce RTX 3090 GPUs. We consistently utilize the AdamW optimizer [50] for model optimization and set the batch size to 64. The initial learning rate is set to 2e-4. In addition, for experiments performed on the HO3D dataset [19], we resize the images to $512 \times 512$

**Table 1: Ablation study on each key component of LCP. †denotes using the CFM backbone.**

| Methods | Components | | | | Hand | | Object |
| | w Context | w Mask | w Sharing | CFM | PA-MPJPE ↓ | MPJPE ↓ | ADD(-S) ↑ (Average) |
|---|---|---|---|---|---|---|---|
| Baseline | - | - | - | - | 5.51 | 13.37 | 47.6 |
| | ✓ | - | - | - | 5.35 | 12.72 | 48.0 |
| LCP | ✓ | ✓ | - | - | 5.37 | 12.66 | 48.3 |
| | ✓ | ✓ | ✓ | - | 5.33 | 12.50 | 49.6 |
| LCP† | ✓ | ✓ | ✓ | ✓ | 5.14 | 11.81 | 50.6 |

**Table 2: More ablation study on the context decoder layer.**

| Model | Hand | | Object |
| | PA-MPJPE ↓ | MPJPE ↓ | ADD(-S) ↑ (Average) |
|---|---|---|---|
| Ours | 5.33 | 12.50 | 49.6 |
| variant 1 | 5.34 | 12.95 | 48.8 |
| variant 2 | 5.35 | 12.84 | 49.5 |

**Table 3: Ablation study on using 2D hand-joint queries.**

| Methods | Hand | | Object |
| | PA-MPJPE ↓ | MPJPE ↓ | ADD(-S) ↑ (Average) |
|---|---|---|---|
| LCP | 5.33 | 12.50 | 49.6 |
| LCP w/o 2D joints | 5.77 | 13.94 | 49.0 |

pixels and set the number of training epochs to 60. Due to the Dex-YCB dataset [3] being significantly larger, we resize all its images to $256 \times 256$ pixels and reduce the number of training epochs to 40. Besides, we set the number of context, hand, and object decoder layers to 1, 3, and 3, respectively.

### 4.3 Ablation Study

We conduct ablation studies on the large-scale Dex-YCB database and employ ResNet-50-FPN as the backbone unless otherwise specified. CFM refers to customized feature maps.

**Effectiveness of key components in LCP.** The experimental results are summarized in Table 1. To ensure a fair comparison with LCP, the baseline model contains four hand and four object decoder layers. These layers use $\mathbf{F}^h$ and $\mathbf{F}^o$ as the value and key in the cross-attention operations, respectively. Then, we replace the first hand and object decoder layers in the baseline model with a context decoder layer, respectively. The two context decoder layers do not share parameters. This experiment is denoted as 'w Context' and significantly reduced the hand's MPJPE by 0.65 mm and PA-MPJPE by 0.16 mm, with a slight performance improvement in the object's ADD(-S) score. This suggests that image-wide context is essential for hand pose estimation.

Next, we introduce a mask to the object branch's context decoder layer, constraining the cross-attention operation in Eq. 2 within the object bounding-box area. This experiment is presented as 'w Mask' in Table 1, and it slightly improves the performance of the object pose estimation, supporting our conjecture that rigid objects may not require extensive contextual information. Furthermore, 'w Sharing' implies sharing the context decoders' parameters with the hand and object. This operation reduces the hand's MPJPE by 0.16 mm and increases the object's ADD(-S) score by 1.3%. These results demonstrate that the interacting hand and object are mutually the most important context for each other.

Finally, with the robust CFM backbone, the performance of LCP improves significantly, reducing the hand's MPJPE by 0.69 mm, PA-MPJPE by 0.19 mm, and increasing the object metric by 1.0%. The above experimental results validate the effectiveness of LCP.

**Ablation study on the context decoder layer.** In Table 2, we compare the performance of LCP's context decoder layer with two potential variants. In the first variant, a mask for $\mathbf{Q}^h$ is imposed on

**Table 4: Ablation study on the CFM backbone.**

| Model | Hand | | Object |
|---|---|---|---|
| | PA-MPJPE ↓ | MPJPE ↓ | ADD(-S) ↑ (*Average*) |
| Ours | 5.14 | 11.81 | 50.6 |
| variant 1 | 5.22 | 11.97 | 49.6 |
| variant 2 | 5.18 | 11.95 | 50.5 |

**Table 5: Results of different numbers of context, hand, and object decoder layers.**

| Context | Hand | Object | PA-MPJPE ↓ | MPJPE ↓ | ADD(-S) ↑ (*Average*) |
|---|---|---|---|---|---|
| 1 | 3 | 3 | 5.33 | 12.50 | 49.6 |
| 1 | 1 | 1 | 5.58 | 13.23 | 42.3 |
| 1 | 5 | 5 | 5.20 | 12.36 | 52.2 |
| 2 | 3 | 3 | 5.32 | 12.52 | 49.8 |
| 3 | 3 | 3 | 5.31 | 12.43 | 50.1 |

the context decoder layer, restricting the cross-attention operation within the hand bounding-box, which is similar to the operation defined in Eq.2. In the second variant, we enlarge the cross-attention operation for $Q^h$ to cover both the hand and object bounding-box areas.

As shown in Table 2, the performance of both variants is lower than our method, especially on the challenging MPJPE metric. Indeed, without procrustes alignment with the ground-truth, the MPJPE metric requires more context for robust hand pose estimation. These experimental results justify the effectiveness of our designs.

**Effectiveness of using 2D hand-joint queries.** Table 3 compares LCP's performance with and without the queries for detecing 2D hand joints. We observe that using the 21 hand-joint queries substantially improves the hand's PA-MPJPE and MPJPE metrics (i.e., 0.44mm and 1.44mm, respectively). This is because 3D hand pose estimation and 2D hand joint prediction are closely related. It also increases the object's ADD(-S) score by 0.6%. This may be because high-quality hand features provide more accurate context for object pose estimation.

**Ablation study on the CFM backbone.** We compare CFM's performance with two possible variants, as displayed in Table 4. The two variants' model structures are provided in the supplementary material. The first variant directly utilizes the disentangled hand and object feature maps produced by the original backbone in [45] for the context decoder layer. In other words, the hand and object queries utilize their respective feature maps in the context layer instead of our combined feature maps. The second variant replaces our concatenation operation described in Section 3.4 with simple element-wise addition between the hand and object feature maps.

As shown in Table 4, the performance of the original backbone model in [45] is significantly lower than ours, which may be because the disentangled feature maps in [45] lose the hand or object context. Moreover, the second variant's performance is only slightly lower than ours, meaning that our proposal can be achieved using the simple element-wise addition or concatenation operations.

**Number of decoder layers.** In Table 5, "Context," "Hand", and "Object" denote the number of context, hand, and object decoder layers, respectively. The table shows that increasing the number

**Table 6: Comparison with state-of-the-art methods on Dex-YCB in terms of hand pose estimation metrics. † denotes using the CFM backbone.**

| | Methods | PA-MPJPE ↓ | MPJPE ↓ |
|---|---|---|---|
| Single-hand | METRO [43] | 6.99 | 15.24 |
| | Spurr et al. [58] | 6.83 | 17.34 |
| | HandOccNet [55] | 5.80 | 14.04 |
| | H2ONet [74] | 5.7 | 14.0 |
| Hand-object | HFLNet [45] | 5.47 | 12.56 |
| | LCP | 5.33 | 12.50 |
| | LCP† | **5.14** | **11.81** |

of context, hand, or object decoder layers improves performance. However, increasing the number of decoder layers also results in more computational costs. To strike a balance between performance and efficiency, we utilize only one context decoder layer and three hand-object decoder layers in the subsequent experiments.

### 4.4 Comparisons with State-of-the-Art Methods

**Comparisons on Dex-YCB.** Dex-YCB [3] is a recently released database. Table 6 shows that LCP outperforms the state-of-the-art 3D hand-object pose estimation methods [45] and surpasses approaches that only perform the hand pose estimation task [55]. Specifically, with the ResNet-50-FPN backbone and the backbone introduced in Section 3.4, LCP's MPJPEs perform better than the state-of-the-art method [45] by 0.06mm and 0.75mm, respectively; and LCP's PA-MPJPEs outperform those of [45] by 0.14mm and 0.33mm, respectively.

Table 7 compares LCP's ADD(-S) scores for object pose estimation with those of state-of-the-art methods. Specifically, with the ResNet-50-FPN backbone and the backbone introduced in Section 3.4, LCP's ADD(-S) score significantly outperforms the state-of-the-art method [45] by 19.4% and 20.4%, respectively. This superiority can be attributed to the advantages of employing disentangled decoder layers in our approach. In particular, our method leverages both robust contextual information and fine-grained regional hand or object features. Simultaneously, the self-attention operations within the decoder facilitate learning the dependencies between object keypoints. Consequently, our approach shows stronger robustness on the challenging Dex-YCB dataset [3] compared to HFLNet [45].

**Comparisons on HO3D.** Table 8 shows that LCP achieves superior performance on hand pose estimation. Specifically, with very similar backbones, LCP† outperforms the recent method [45] by 4.7mm in terms of the MPJPE score. Compared with the PA-MPJPE score, MPJPE measures the mean joint position error without aligning the estimated hand pose with the ground truth; therefore, it may be more practical in real-world applications. We attribute the advantages of LCP† to the exploration of image-wide context, which facilitates holistic estimation of hand poses. Moreover, LCP achieves comparable performance even when compared with approaches that only focus on hand pose estimation (i.e., the "single-hand" approaches), as shown in Table 8.

Table 9 further compares the object pose estimation accuracy between our method and state-of-the-art approaches. Specifically, equipped with similar backbones, the average ADD score of LCP†

**Table 7: Comparisons with state-of-the-art method [45] on Dex-YCB in terms of object pose estimation metrics.**

| Method | Lin et al. [45] | LCP | LCP$^\dagger$ |
|---|---|---|---|
| Metric | ADD(-S) ↑ | ADD(-S) ↑ | ADD(-S) ↑ |
| master_chef_can | 23.3 | 50.6 | **50.8** |
| cracker_box | 66.6 | **92.9** | 91.6 |
| sugar_box | 35.6 | **70.1** | 67.9 |
| tomato_soup_can | 12.2 | 26.4 | **30.6** |
| mustard_bottle | 48.1 | 69.3 | **70.8** |
| tuna_fish_can | 8.6 | 19.6 | **21.8** |
| pudding_box | 31.2 | 56.9 | **57.8** |
| gelatin_box | 26.0 | 48.1 | **50.2** |
| potted_meat_can | 21.1 | 37.4 | **41.1** |
| banana | 16.9 | **35.1** | 33.3 |
| pitcher_base | 36.5 | 60.3 | **66.6** |
| bleach_cleanser | 42.5 | 67.0 | **70.0** |
| bowl* | 36.2 | 53.0 | **55.0** |
| mug | 16.8 | **31.0** | 27.6 |
| power_drill | 45.1 | 70.8 | **76.3** |
| wood_block* | 45.9 | **64.8** | 63.9 |
| scissors | 13.6 | **28.7** | 28.3 |
| large_marker | 3.7 | **9.5** | 8.6 |
| extra_large_clamp* | 44.8 | 52.0 | **52.5** |
| foam_brick* | 28.8 | **48.0** | 47.1 |
| average | 30.2 | 49.6 | **50.6** |

**Table 8: Comparisons with state-of-the-art methods on HO3D in terms of hand pose estimation metrics. '(P)' denotes pre-training on the Dex-YCB dataset.**

| | Methods | PA-MPJPE ↓ | MPJPE ↓ |
|---|---|---|---|
| Single-hand | Pose2Mesh [10] | 12.5 | 33.3 |
| | Hasson et al. [24] | 11.0 | 31.8 |
| | I2L-MeshNet [53] | 11.2 | 26.0 |
| | Hampali et al. [19] | 10.7 | 30.4 |
| | METRO [43] | 10.4 | 28.9 |
| | H2ONet [74] | 9.0 | - |
| | HandOccNet [55] | 9.1 | 24.0 |
| Hand-object | Hasson et al. [23] | 11.4 | 36.9 |
| | Liu et al. [47] | 10.1 | 31.7 |
| | ArtiBoost [40] | 11.4 | 25.3 |
| | Keypoint Trans [20] | 10.8 | 25.7 |
| | HFLNet [45] | 8.9 | 28.4 |
| | HFLNet (P) [45] | 8.7 | 27.0 |
| | LCP$^\dagger$ | 8.9 | 23.7 |
| | LCP$^\dagger$ (P) | **8.5** | **21.5** |

is higher than [45] by 9.1%. The above comparisons validate the effectiveness of our approach.

Finally, since the HO3D dataset is relatively small, overfitting may occur on this database. To cope with this problem, we conduct an additional experiment that is similar to the one in H2ONet [74].

**Table 9: Comparisons with state-of-the-art methods on HO3D in terms of object pose estimation metrics. '(P)' denotes pre-training on the Dex-YCB dataset.**

| Methods | cleanser ↑ | bottle ↑ | can ↑ | average ↑ (ADD) |
|---|---|---|---|---|
| Liu et al. [47] | 88.1 | 61.9 | 53.0 | 67.7 |
| HFLNet [45] | 81.4 | 87.5 | 52.2 | 73.3 |
| HFLNet (P) [45] | 91.9 | 77.0 | 59.4 | 76.1 |
| LCP$^\dagger$ | **95.4** | 92.0 | 60.0 | 82.4 |
| LCP$^\dagger$ (P) | 93.8 | **92.9** | **69.5** | **85.4** |

**Table 10: Comparisons with state-of-the-art methods on InterHand2.6M. $^\diamond$ denotes the LCP variant without using ground-truth hand bounding boxes.**

| | MPJPE | MPVPE | MPJPE-S | MPVPE-S |
|---|---|---|---|---|
| Moon et al. [54] | 16.02 | - | - | - |
| Fan et al. [13] | 14.27 | - | - | - |
| Zhang et al. [82] | - | - | 11.28 | 12.01 |
| IntagHand [42] | 10.27 | 10.53 | 9.40 | 9.68 |
| ACR [81] | 9.08 | 9.31 | 8.41 | 8.53 |
| LCP$^\diamond$ | 8.39 | 8.74 | 7.41 | 7.63 |
| LCP | **8.09** | **8.46** | **7.10** | **7.33** |

Specifically, we first pre-train LCP$^\dagger$ and HFLNet [45] on the large-scale Dex-YCB database, respectively. Then, we fine-tune each of them on the HO3D dataset. Experimental results show that with proper pre-training, LCP$^\dagger$ outperforms HFLNet with very similar backbones.

**Comparisons on InterHand2.6M.** We extend our approach to the 3D interacting-hands pose estimation task [13, 42, 54, 81, 82]. Specifically, we utilize LCP's hand and object decoder layers to predict the left- and right-hand poses, respectively. In this extension, we remove the masking operation in Eq. 2, enabling the right-hand to search for image-wide context. In Table 10, we compare LCP's performance with state-of-the-art methods on the InterHand2.6M database [54]. Similar to ACR [81], we evaluate our method using metrics such as MPJPE and Mean Per Vertex Position Error (MPVPE), as well as their scaled versions denoted as MPJPE-S and MPVPE-S, respectively.

Table 10 shows that LCP continues to achieve the best performance across all metrics: MPJPE, MPVPE, MPJPE-S, and MPVPE-S. Moreover, existing 3D interacting-hands pose estimation methods usually do not assume the availability of hand bounding boxes. Therefore, to facilitate fair comparison with these approaches, we adopt the bounding box enclosing all 2D joints of a hand that are predicted by the context decoder layer as the bounding box for this hand. To promote hand joint prediction accuracy, we stack one more context decoder layer and employ the second context layer for hand joint prediction. We denote this LCP variant as LCP$^\diamond$ in Table 10. Furthermore, Table 10 shows that LCP$^\diamond$ still outperforms state-of-the-art methods. Thus, the experimental results validate LCP's effectiveness.

## 4.5 Qualitative Comparisons

Fig. 4 illustrates the qualitative comparisons between LCP$^\dagger$ and the state-of-the-art methods [45]. We compare their performance on

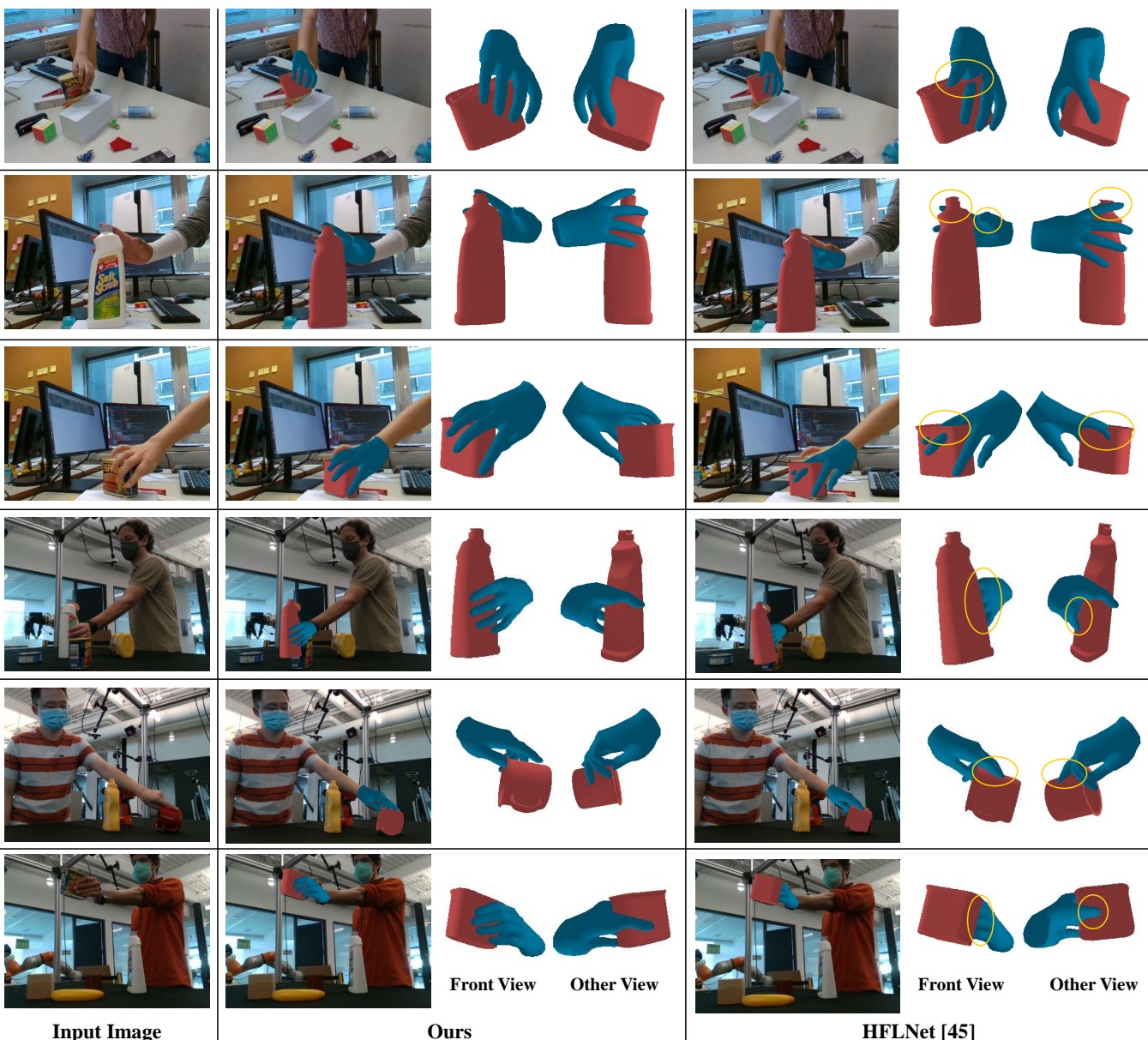

**Figure 4: Qualitative comparisons between LCP[†] and state-of-the-art methods [45] on the HO3D [19] (the first three rows) and Dex-YCB [3] (the remaining rows) databases.**

images with severe hand-object occlusions. It is shown that LCP[†] estimates the hand and object poses more robustly. Since LCP[†] and [45] adopt similar backbones, we attribute the advantages of LCP[†] to its robust learning capacity of broad-ranged context and fine-grained regional features, facilitating the accurate estimation of hand and object poses.

## 5  CONCLUSION AND LIMITATIONS

This paper explores robustly achieving a broad context range for the 3D hand-object pose estimation task. Our approach stacks disentangled transformer decoder layers to extract image-wide context,

hand, and object regional features. By imposing priors to the context decoder layer, our model robustly extracts context for the hand and object, respectively. We also introduce customized feature maps for the three decoder layer types Finally, our approach outperforms existing methods on 3D hand-object and interacting-hands pose estimation. However, this study encounters certain limitations. For example, the utilized LCP queries remain the same for different images. In the future, we will enable adaptive adjustment of each image's queries according to its content to enhance pose estimation accuracy.

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
