# OpenReview forum: "Learning Context with Priors for 3D Interacting Hand-Object Pose Estimation"
_acmmm.org/ACMMM/2024/Conference — MM2024 Poster_

### Official Review · Reviewer_7CjX · 2024-05-16

**Rating:** 5
**Confidence:** 3

**Summary:**

Achieving 3D hand-object pose estimation in interaction scenarios is challenging due to severe occlusions that occur during the interaction process. Existing methods address this issue by leveraging the correlation between hand and object poses as additional cues. These methods overlook the role of context and instead use stacked transformer decoder layers to extract image-wide context and regional hand or object features. By sharing the parameters of the context decoder layer between hand and object pose estimation tasks, the prior knowledge that hands and objects are the most important context for each other is reinforced. Since the context, hand, and object decoder layers play different roles, the article provides customized feature maps for these layers to facilitate decoupling, enabling the model to more accurately learn and predict the poses of hands and objects. Finally, experiments were conducted on several different datasets, and the results demonstrate the superiority of this method.

**Strengths:**

1. By stacking transformer decoder layers and limiting the scope of cross-attention, the method extracts image-wide context as well as regional hand and object features.
2. The robustness of context features is improved by sharing the parameters of the context decoder layer, emphasizing the interrelation between hands and objects.
3. Customized feature maps are provided for the context, hand, and object layers, which helps in decoupling these layers while ensuring they operate within a similar feature space.
4. Experiments on the HO3D and Dex-YCB datasets demonstrate that this method has improved performance compared to other methods.

**Limitations:**

1. The paper mentions that the LCP framework uses the same queries for different images. This means that the model cannot adaptively adjust the queries based on the specific content of the image, which may affect the accuracy of the pose estimation for certain images. Future work will consider adaptively adjusting the queries according to the image content to improve the accuracy of pose estimation.
2. While the introduction of customized feature maps and the shared context decoder layer improves performance, it also increases the complexity and computational load of the model.
3. Although increasing the number of decoder layers can enhance performance, it also increases computational costs. In practical applications, a trade-off may need to be made between performance and efficiency.
4. The paper does not discuss the real-time performance of the model, which is quite important for many practical application scenarios, such as augmented reality, virtual reality, or real-time robotic manipulation.

**Suitability:**

3

---

### Official Review · Reviewer_qJCv · 2024-05-24

**Rating:** 5
**Confidence:** 4

**Summary:**

The authors propose a novel network to for joint hand-object pose estimation from a single image.  To leverage a broad range of the image contexts, they leverage the stacked transformer decoders layers to extract the features. This learned a hand-object interaction prior and reducing the feature learned complexity by disentangle hand and object. Their method is evaluated on the HO3D and DexYCB dataset, and the results significantly outperforms state-of-the-art approaches.

**Strengths:**

1.	The proposed pipeline utilized the transformer decoder layer to model the hand-object interactions and extract the robust features.
2.	The paper conducts comprehensive evaluation and ablation studies.

**Limitations:**

1.	Insufficient qualitative attention maps and their comparative analysis. The reasoning behind the network design remains unclear, and the insufficient provision of attention maps makes it challenging to assess the method's effectiveness.
2.	Lin et al. [34] exhibit significantly better performance than LCP in terms of PA-MPJPE on the HO3D dataset. Could the authors explain the reasons behind this observation?
3.	In comparison with Lin et al. [34], what contributes to the improvement in object pose estimation in this method? Given that Lin et al. employ a similar network architecture to LCP, an explanation for the observed enhancement would be valuable.
4.	The authors did not provide information about the running speed.
5.	It would be good to analyze failure cases to better understand the limitations of the proposed system.

**Suitability:**

3

---

### Official Review · Reviewer_qjmn · 2024-05-24

**Rating:** 4
**Confidence:** 4

**Summary:**

This paper addresses the challenge of 3D hand-object pose estimation in interaction scenarios, particularly focusing on the severe occlusion that occurs during interactions. Traditional methods leverage the correlation between hand and object poses as additional cues by extracting features from their respective regions and refining them mutually. However, these methods often overlook the importance of broader image context. To tackle this issue, the paper proposes a novel and robust approach that incorporates priors to learn a wide range of contextual information. Specifically, the approach uses stacked transformer decoder layers to extract both image-wide context and regional hand or object features by constraining cross-attention operations. The parameters of the context decoder layers are shared between hand and object pose estimations to avoid interference in the context-learning process. This imposes a prior that the hand and object are mutually the most important contexts for each other, significantly enhancing the robustness of the context features. Additionally, customized feature maps are provided for the context, hand, and object decoder layers, facilitating disentanglement and reducing feature learning complexity.

**Strengths:**

1. Stacked Transformer Decoder Layers: The approach uses stacked transformer decoder layers to extract image-wide context and fine-grained features by restricting the scope of cross-attention operations, enabling detailed feature extraction.
2. Shared Context Decoder Layer Parameters: By sharing the context decoder layer parameters between hand and object pose estimation tasks, the method imposes a prior that the hand and object are the most important contexts for each other. This enhances the robustness of the extracted context features.
3. Customized Feature Maps: The method provides customized feature maps for the context, hand, and object decoder layers. It utilizes an efficient 1x1 convolution layer to concatenate the hand and object feature maps along the channel dimension and halve the number of channels, significantly improving the performance of 3D hand-object pose estimation.

**Limitations:**

1. Insufficient Motivation: The motivation for the paper is inadequate. It does not clearly explain the benefits and necessity of using a broad range of image context. Additionally, there are no ablation studies to verify the advantages of incorporating this context.
2. Lack of Novelty: The core idea of the paper lacks originality, with limited innovation. The main modules, such as the stacked transformer decoder layers and context decoder layers, are not original to this paper. The LCP module is merely a targeted improvement based on the work presented in [45].
3. Insufficient Experimental Support: The experimental results are insufficient to support some of the claims made in the paper. There is a lack of comparison with both the latest and most classical methods. Tables 6 and 7 only provide comparisons with the method in [45], which is inadequate. Additionally, it is recommended to include the corresponding computational costs in Figure 5 to support the related claims in the paper.
4. Clarity and Organization: The clarity and organizational structure of the manuscript need improvement, and the overall writing quality should be enhanced. It is suggested to include a clear summary of contributions at the end of the Introduction to help readers understand the key points.
5. Definition of "Impose Priors": The term "impose priors in the context-learning process" is mentioned multiple times without a clear definition. It is recommended to provide a precise explanation of this term.
6. Explanation of Learnable Queries: In line 336, it is not clear how "one group of learnable queries Qh and Qo" is obtained. Please provide an explanation.
7. Definition of Position Encoding: Provide a formula or definition for the position encoding Epos mentioned in line 352.
8. Justification of Operations: In lines 352-355, explain the operations performed and why they are expected to achieve the desired outcomes as stated by the authors.

**Suitability:**

3

---

### Official Review · Reviewer_FW9Q · 2024-05-31

**Rating:** 3
**Confidence:** 4

**Summary:**

To better model hand-object interaction, unlike most existing methods which only use the hand-object correlation as additional cues, the paper proposes to impose priors during the learning process so that the model could take full advantage of both hand and object context information, leading to more robust context features.

The pipeline contains a context decoder along with hand and object queries for imposing the proposed interaction priors, and then hand and object decoder would be used for respective pose estimation.

The experimental results on HO3D, Dex-YCB are shown superior to previsou methods.

**Strengths:**

- The paper is easy to follow
- The experimental results for hand/object pose estimation show the effectiveness of the proposed pipeline compared with previous methods.

**Limitations:**

- The proposed idea: Using attention layer to capture correlation between hand and object and then condition respective pose decoder on such learned correlation is incremental to the hand-object interaction community.
- How does the detection performance affect the context learning and final pose estimation?
- Table 9 shows a worse performance on cleanser after pre-training, could the author offer some insights?
- Could the authors provide results showing the generalization aiblity of the proposed method?
- Does MPJPE adopted in the paper mean calculating the error in camera space? If so, as shown in both Table 6 and Table 8, could the authors elaborate more on why MPJPE shows more performance gain than PA-MPJPE?

**Suitability:**

3

---

### Meta-Review · Area_Chair_U27k · 2024-07-07

**Recommendation:** Accept (Poster)
**Confidence:** 5

**Metareview:**

This paper initially received 1 reject, but the reviewer revised their opinion. The AC agree with the reviewers' comments.